# Facial asymmetry and midsagittal plane definition in 3D: A bias-free, automated method

**Nikolaos Gkantidis**[1]ⓘ*, **Jasmina Opacic**[1]◉, **Georgios Kanavakis**[2], **Christos Katsaros**[1], **Demetrios Halazonetis**ⓘ[3]

1 Department of Orthodontics and Dentofacial Orthopedics, School of Dental Medicine, University of Bern, Bern, Switzerland, 2 Department of Orthodontics and Pediatric Dentistry, UZB–University School of Dental Medicine, University of Basel, Basel, Switzerland, 3 Department of Orthodontics, School of Dentistry, National and Kapodistrian University of Athens, Athens, Greece

◉ These authors contributed equally to this work.
* nikolaos.gkantidis@unibe.ch

## Abstract

Symmetry is a fundamental biological concept in all living organisms. It is related to a variety of physical and social traits ranging from genetic background integrity and developmental stability to the perception of physical appearance. Within this context, the study of human facial asymmetry carries a unique significance. Here, we validated an efficient method to assess 3D facial surface symmetry by best-fit approximating the original surface to its mirrored one. Following this step, the midsagittal plane of the face was automatically defined at the midpoints of the contralateral corresponding vertices of the superimposed models and colour coded distance maps were constructed. The method was tested by two operators using facial models of different surface size. The results show that the midsagittal plane definition was highly reproducible (maximum error < 0.1 mm or˚) and remained robust for different extents of the facial surface model. The symmetry assessments were valid (differences between corresponding bilateral measurement areas < 0.1 mm), highly reproducible (error < 0.01 mm), and were modified by the extent of the initial surface model. The present landmark-free, automated method to assess facial asymmetry and define the midsagittal plane of the face is accurate, objective, easily applicable, comprehensible and cost effective.

## Introduction

Asymmetry of biological organisms is a critical concept for several principles providing valuable cues, from genetic background integrity and developmental stability to the perception of physical appearance [1–3]. Thus, thorough assessment of the morphological asymmetry of a structure is crucial for many different disciplines and has various applications.

Three types of asymmetry have been identified in biological structures: fluctuating asymmetry, directional asymmetry and antisymmetry. Fluctuating asymmetry describes the small left-right differences due to random imprecisions in development. Directional asymmetry is the

**Data Availability Statement:** All relevant data are within the paper and its Supporting Information files, including the anonymized datasets generated and analyzed during the current study (S1 Dataset). Due to the sensitive nature of the surface models

used in this study, which contain potentially identifying patient information, participants were assured raw data would remain confidential and would not be shared. Data requests may be sent to the Swiss Ethics Committee of the Canton of Bern (email: info.kek@be.ch).

**Funding:** This work was partially supported by the FLAG-ERA grant (JTC 2019 project MARGO) and the Greek General Secretariat for Research and Technology (GSRT, grant number: T11ERA4-00017) awarded to author DH. The specific roles of this author are articulated in the 'author contributions' section. The funders had no role in the study design, data collection and analysis, decision to publish, or preparation of the manuscript.

**Competing interests:** I have read the journal's policy and the authors of this manuscript have the following competing interests: Demetrios Halazonetis owns stock in dHAL Software, the company that markets Viewbox 4. This does not alter our adherence to PLOS ONE policies on sharing data and materials. Demetrios Halazonetis was not involved in data generation and analysis, and thus, could not affect the study outcomes. All other authors declare no known competing financial interests or personal relationships that could have appeared to influence the work reported in this paper.

tendency for a trait on the left or right body side, for example the arrangement of internal organs. Antisymmetry describes the direction of the asymmetries resulting in a sum of "left-sided" and "right-sided" individuals [4, 5].

So far, several ways have been suggested to assess morphological asymmetry at a single time point [6–8]. Providing that adequate methods are available for this, the extent and location of asymmetry in a living organism can be followed over time by repeating the process in subsequent documentations [9]. In most conventional techniques, the first and critical step involves bilateral landmark identification either for direct measurements or for the construction of a midsagittal plane that delimits the two contralateral sides. Other commonly used techniques require direct landmark identification on the midsagittal plane of an object to define it [5, 10, 11]. These landmarks are usually very few. Regardless of the method used, the step of identifying landmarks by the investigator affects the validity of the process [12–14].

Geometric morphometric approaches have been developed to overcome the limitation of the midsagittal plane construction based on an arbitrarily selected limited number of landmarks. With this approach, the anatomical form of a structure is captured as a landmark configuration. The average shape (consensus) following the Procrustes fit of such a configuration and its mirrored configuration produces a perfectly symmetric shape [5, 15, 16]. Thus, the middle of all lines that connect the corresponding right and left side landmarks of this totally symmetric configuration (consensus) will be located on a single line (for 2D, or plane for 3D), which will constitute the midsagittal plane of the original and the mirrored configuration. This approach is more advantageous than others because it estimates asymmetry based on the entire landmark configuration and avoids assumptions that place more weight on certain landmarks that might be considered more stable or lying on an arbitrarily defined middle [5]. However, the power of this approach still lies on the identification of a considerable number of landmarks, which is a time consuming and error prone process [17]. Landmark identification on 3D objects is usually even more complex and time consuming [18–20]. Finally, this procedure assumes that identifiable landmarks are available. Thus, it cannot be used on smooth surfaces without landmarks.

Due to its prominent position and role in the human body, the human face has attracted increased interest also in this field [20–22]. For example, asymmetric faces are perceived as less attractive, and this can affect several aspects of personal, social, and professional life [2, 22]. Thus, the human face is a typical example where all the above methods have been repeatedly applied by various disciplines. As in any biological organism, there is no human face that is perfectly symmetric. However, the extent and location of facial asymmetries differs considerably among individuals. Apart from research purposes, thorough assessment of asymmetry is crucial in clinical practice for diagnosis, treatment planning, and outcome assessment [23–26].

Recent tools allow fast and reliable generation of 3D facial surface models [27–29], which might be used to assess facial asymmetry in three dimensions, avoiding the need for placement of anatomical landmarks. Here, we suggest and validate a novel method for automated midsagittal plane construction and facial asymmetry assessment, which is efficient, highly informative and not prone to processing- or operator-related errors. The effect of differences in the extent of the tested 3D surface on the outcomes was also investigated.

## Materials and methods

### Ethical approval

This project describes a prospective methodological study using retrospectively obtained diagnostic data and has been approved by the Swiss Ethics Committee of the Canton of Bern (Protocol No. 2019–01815, Date of Approval: 17.12.2019). The first two authors accessed the

respective archives for research purposes on 15.01.2020 to 05.06.2020 and had access to information that could identify individual participants during data collection. All methods were performed in accordance with the relevant guidelines and regulations. Written informed consent was obtained from all subjects and/or their legal guardian(s) allowing the use of their data for research purposes, as well as the publication of identifying information/images in an online open-access publication, if applicable.

## Sample

The sample consisted of 20 randomly selected 3D photos at rest (ten 8-12-year-old children and ten 20-40-year-old adults, with equal gender distribution), from a preexisting orthodontic patient sample. These photographs were originally obtained in the context of standard pretreatment records of patients treated in the Department of Orthodontics and Dentofacial Orthopedics, University of Bern, using a 3dMD stereophotogrammetric camera (3dMDface system, 3dMD Inc., Atlanta, GA, USA). The 3D photos were obtained with the patient positioned in a standard manner at a distance of 1.2 m from the camera. Background light was removed according to manufacturer's instructions. The machine was calibrated every morning prior to the acquisition of the first image of the day and also when an error message was received.

## 3D model preparation

All image processing was performed in Viewbox 4 software (Version 4.1.0.1 BETA 64, dHAL Software, Kifissia, Greece). The original TSB files obtained during image acquisition were imported in Viewbox 4 software and transformed to STL files, which were used for further processing.

Initially, for visualization purposes only, the 3D photos were oriented with the Frankfurt Horizontal plane (FH) parallel to the floor and cropped as shown in Fig 1. At this stage, two rectangular areas of interest were selected on the original 3D photos: Area A extended vertically from the middle of the eyebrows to a horizontal line passing through the left and right stomia and laterally to the middle of the eye and Area B extended from a line connecting the left inner and outer canthus to the subnasale horizontal level (Fig 1).

Afterwards, the duplicated 3D surface models with the areas of interest selected on them were further cropped as described below. In all cases, the upper limit of the cropping plane was defined at Trichion and the lower, 1 cm dorsal to Menton. Additionally, the duplicate 3D surfaces were cropped posteriorly at: just behind the ear (Crop 1), 2 cm behind the outer canthus (Crop 2), or at one side 2 cm behind the outer canthus and at the other just behind the ear (Crop 3). This was performed using the box tool, with its planes parallel and vertical to FH. A fourth peripheral cropping (Crop 4) was performed on the 10 subjects (5 children and 5 adults) that were used for reproducibility testing, using the manual selection tools (Fig 2). The various croppings enabled the testing of the effect of extend of the facial surface and of the bilateral presence of corresponding anatomical structures on the assessment of asymmetry and the midsagittal plane construction.

## 3D model superimposition and midsagittal plane construction

Each one of the originally cropped models described above (Crop 1) was duplicated and mirrored, including the Area A and Area B selections, so that the measurement areas were selected only once per model, and thus, were always identical on the surface meshes. This allows for direct comparisons between asymmetry outcomes on Areas A and B.

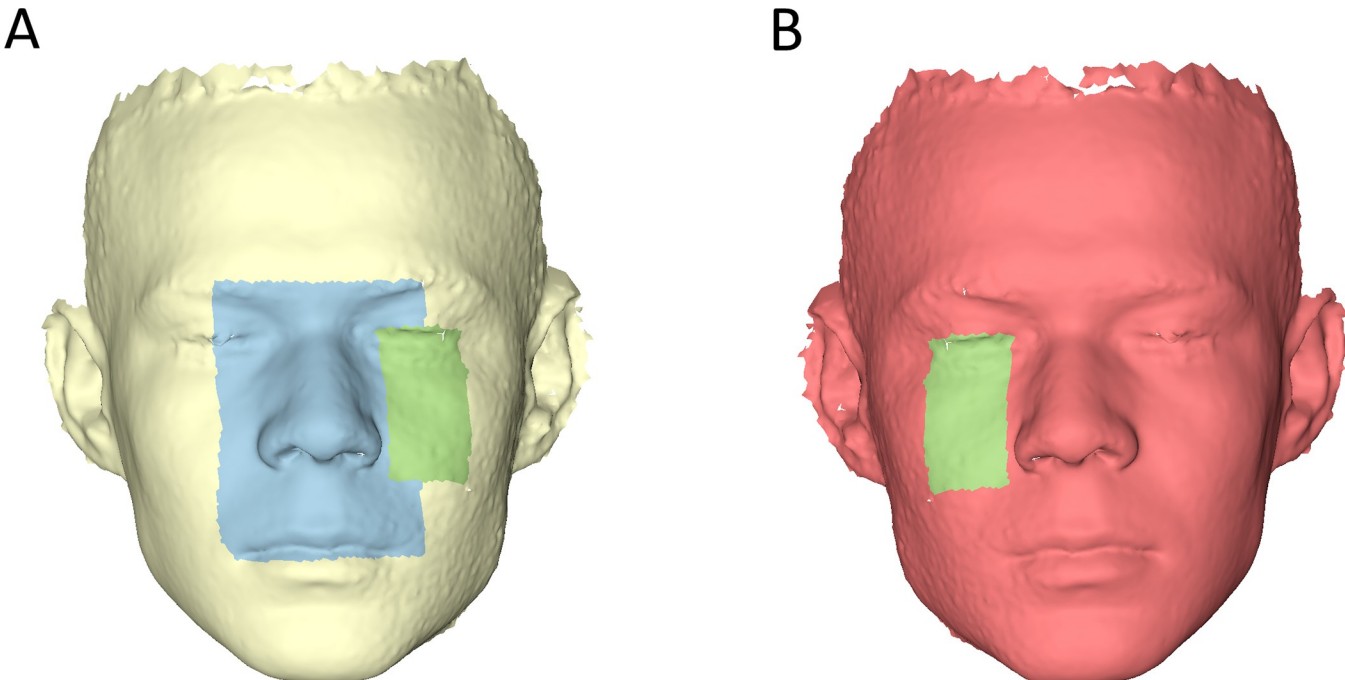

**Fig 1. Areas of interest used to measure distances between original and mirrored superimposed models.** (A) Original 3D facial surface model showing measurement Area A in light blue and Area B right in light green colour. (B) Mirrored copy of the original model, which provided Area B left (light green, mirrored area B right). The original model was mirrored with area B selected, so that the subsequent area B left was identical to area B right, but located on the other side of the face.

The vertices of each mesh are kept in an array, so points of the original and mirrored mesh that have the same array index are corresponding contralateral vertices. It can be proven geometrically that the midpoints of the lines that join corresponding vertices lie on a single plane, independent of the orientation of the two meshes towards each other (Fig 3). This plane can be considered the midsagittal plane, provided that the original and mirrored surfaces have been appropriately approximated. Thus, for all Crops, each initial model was best-fit approximated with its mirrored duplicate following the application of the software's iterative closest point (ICP) algorithm [30], using the following settings: 100% estimated overlap of meshes,

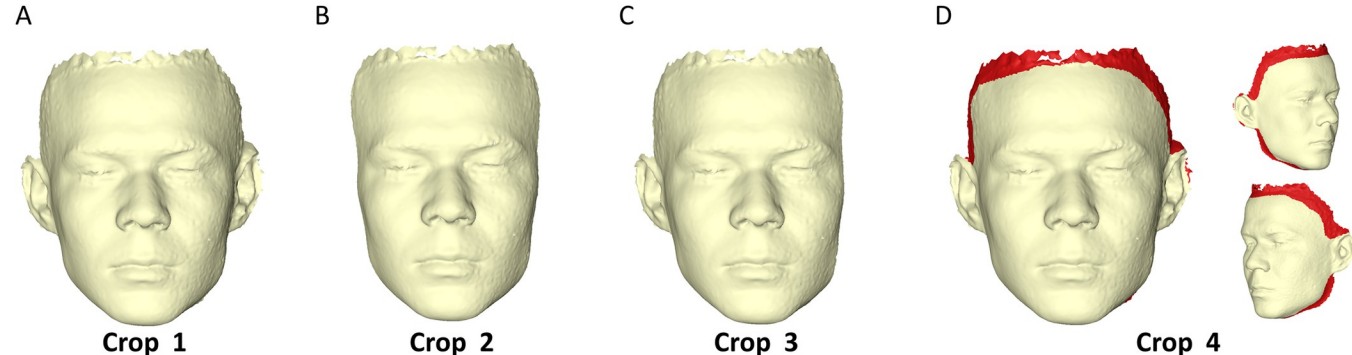

**Fig 2. Differently cropped facial surface models.** (A) Original 3D facial surface model. (B) Duplicated original facial surface model cropped bilaterally 2 cm behind the outer canthus points. (C) Duplicated original facial surface model cropped unilaterally (left side) 2 cm behind the outer canthus point. (D) Duplicated original facial surface model cropped slightly, at random sites, at the periphery of the original surface.

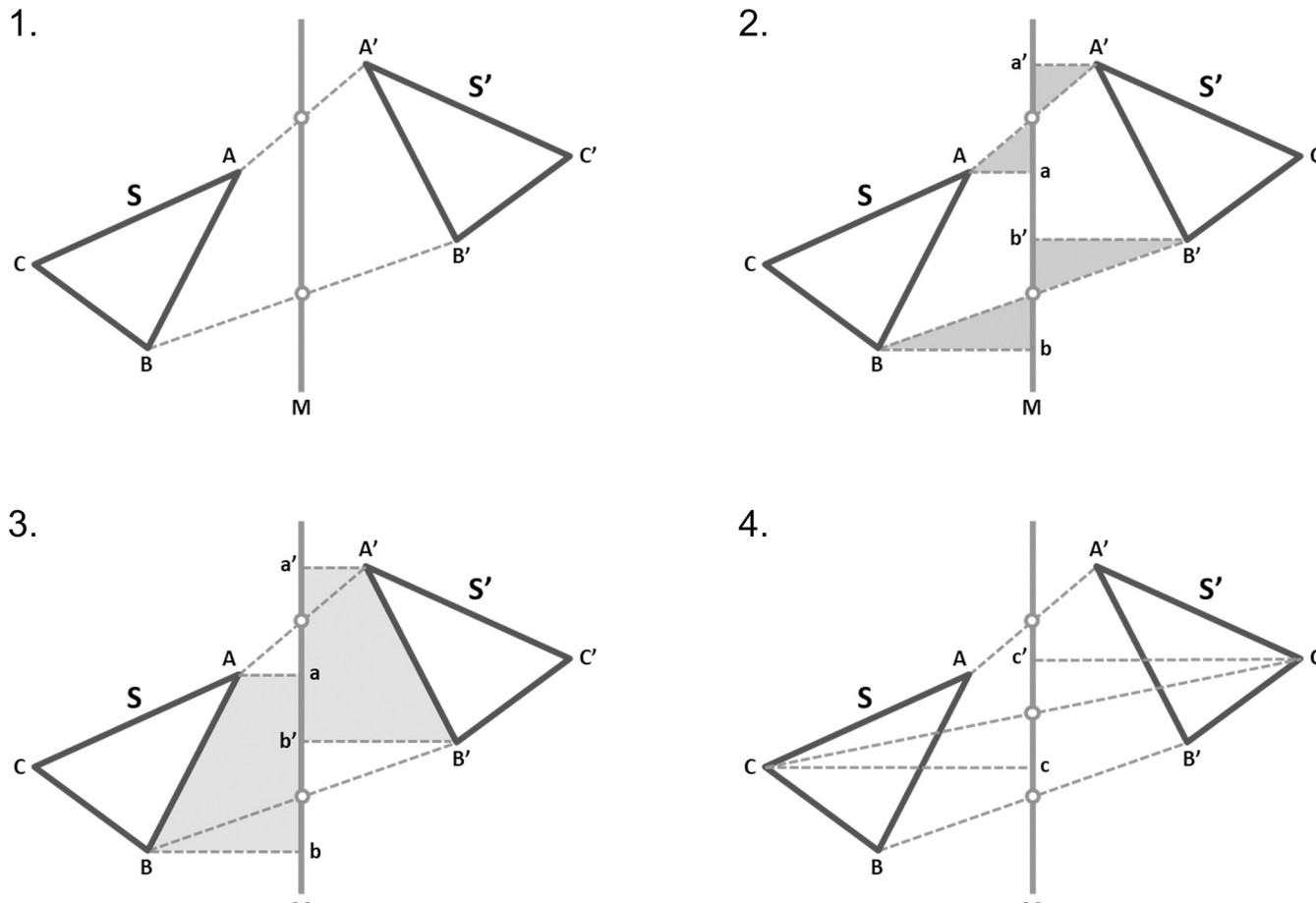

**Fig 3. Proof that the midpoints of the lines that join corresponding vertices always lie on a single plane.** Assume a shape S and its flipped copy S'. 1: Construct line M, passing through the midpoints of the lines joining any two corresponding point pairs, e.g. A—A' and B—B'. 2: Drop perpendiculars from points A, A', B, B' on line M. Resulting triangles are equal, therefore lengths Aa, A'a' and Bb, B'b' are equal, respectively. 3: It follows that quadrilaterals AabB and A'a'b'B' are equal, and shapes S and S' are equally inclined to line M, at the same distance from it. 4. Therefore, any point (e.g. C) is at an equal distance from line M as its corresponding point (C'), and the connecting line (CC') crosses line M at its midpoint. A similar argument holds for the 3D case, where all midpoints of the lines connecting corresponding points are co-planar.

matching point to plane, exact nearest neighbor search, 100% point sampling, and 'exclude overhang regions' option. The original model always remained in its initial position and the mirrored one was approximated to it (Fig 4A). Following the best-fit alignment of the initial and the corresponding mirrored model, the software constructed the midsagittal plane by calculating the plane passing through the midpoints of all lines connecting the corresponding vertices, now located at contralateral sides of the two models (Fig 4B). Although the plane can be computed from just three non-collinear midpoints, for numerical accuracy, the software uses a principal component analysis (PCA) on the coordinates of several midpoints to compute the eigenvector with the smallest eigenvalue. The processing of an original 3D facial photo and the full application of the method in actual conditions is shown in S1 Video.

## Validity

Validity was assessed qualitatively through the visualization of respective colour-coded distance maps by two operators (J.O. and N.G.) independently, through observation of a

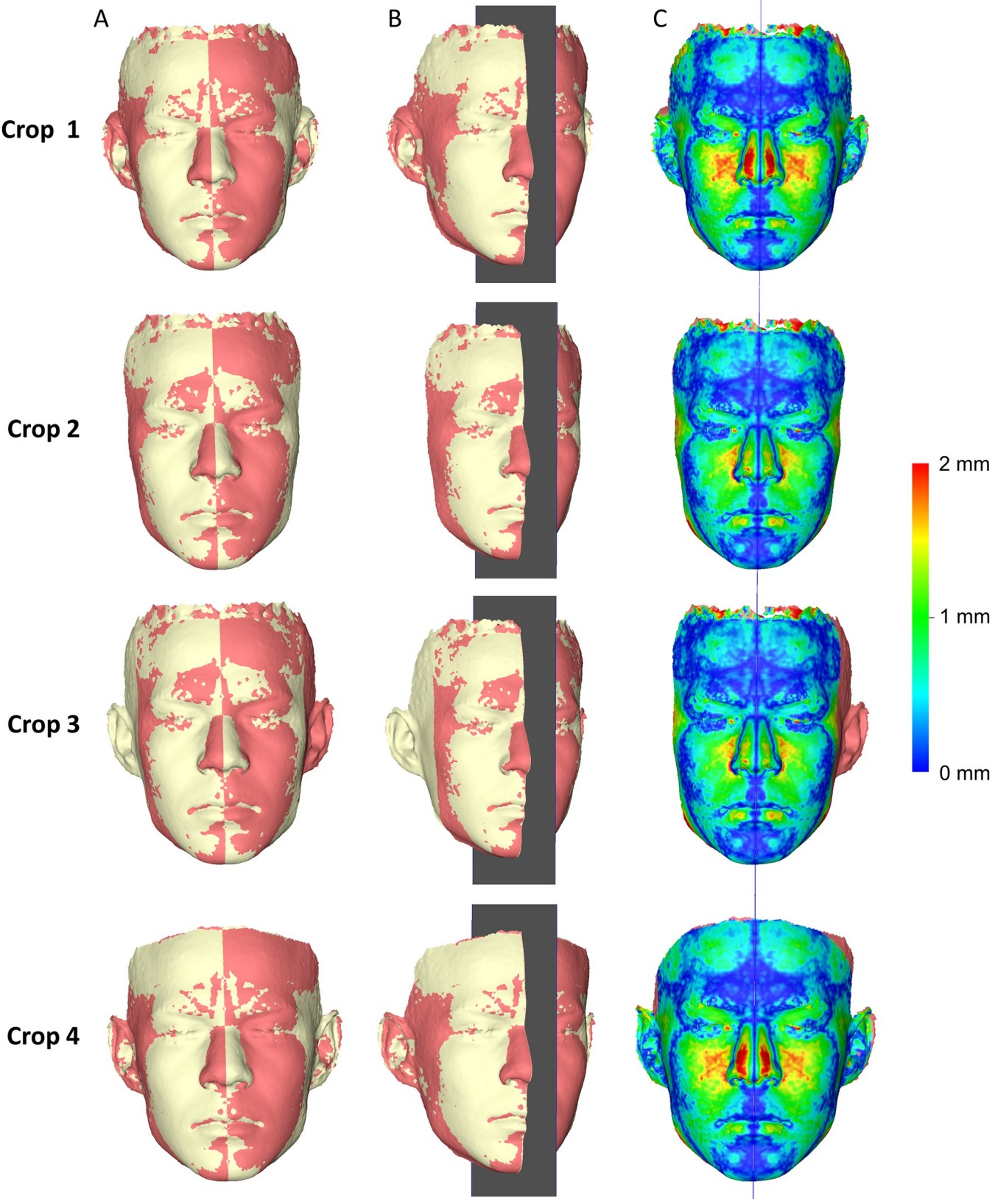

**Fig 4. 3D model superimposition, midsagittal plane construction, and asymmetry assessment.** (A) Differently cropped facial surface models best fit approximated with their mirrored duplicates. (B) Automated midsagittal plane generation. (C) Facial asymmetry indicated by colour-coded distance maps, which show similar distances between the superimposed models at the two sides (right/left).

symmetrical colour pattern in relation to the midsagittal plane (Fig 4C). Furthermore, in case of a perfect approximation of the initial with its corresponding mirrored model, the mean absolute distance (MAD) values between the two models at areas B right and left should be identical, and thus the respective differences should be zero. The further the deviation from zero, the less accurate the superimposition of the two models. Thus, the respective distances were calculated as described above and compared. The robustness of the process, concerning the cropping of the original facial surface model, was also assessed through visualization of colour-coded distance maps. The latter should show similar distances of the superimposed models, at the two sides (right/left), and between different croppings. The respective differences in distances at areas A and B between differently cropped 3D facial models were also compared. There, zero differences would indicate absence of effect of different croppings on the tested outcomes.

## Superimposition outcome reproducibility

To assess intra- and inter-operator reproducibility, two operators repeated all processes in 10 randomly selected samples (5 children and 5 adults), apart from measurement area selection and croppings 1, 2, and 3 that were performed only once to allow for sole assessment of the superimposition process error. On the contrary, cropping 4 was selected each time individually.

For reproducibility testing, the mirrored models with the selected measurement areas were duplicated, their original position in space was changed, and they were again superimposed on the initial models as described above. The MADs between the repeatedly superimposed mirrored models on the original model, in Area A, Area B right and Area B left (Fig 1) were calculated. The sum of these MADs for each single case indicated the amount of error. Ideally, the repeatedly superimposed mirrored models should show zero distance. Furthermore, the cases that showed the maximum error were assessed through visualization of relevant colour-coded distance maps.

To assess the reproducibility of the midsagittal plane construction, the movements (right/left) and the respective rotations required to perfectly match repeatedly created midsagittal planes, through a best-fit superimposition, were calculated. Zero movements and rotations between repeatedly created midsagittal planes would indicate perfect reproducibility.

## Statistical analysis

Data were tested for normality using Shapiro-Wilk test and significant deviations were detected. Thus, non-parametric statistics were applied.

Differences in the distances between the initial and the mirrored model at the measurement areas among different croppings were tested through Friedman's test. In case of significant differences, pairwise comparisons were performed using Wilcoxon signed rank test. In the latter case, a Bonferroni correction was applied to the level of significance to reduce the chances for false positive outcomes. The same approach was applied to test differences in the midsagittal plane, differences between the right and left corresponding measurement areas, as well as systematic differences between repeated superimpositions and midsagittal plane creations. Two-sided tests with alpha level of 0.05 were originally performed. One-sample t-test was applied to test if the differences of repeated measurements deviated significantly from zero.

The effect of differences between operators in superimposition outcomes on the resulting midsagittal planes was tested through Spearman's correlations. For this, differences of Operator 1 from Operator 2 in the MAD of the initial from the mirrored model, at all measurement areas, were correlated to the differences in the generated midsagittal planes.

## Results

### Effect of differential facial surface model extent on asymmetry assessment

There were significant differences in the MAD of all measurement areas between the differently cropped facial models that were approximated through best fit (Friedman test, $p = 0.001$). Pairwise tests between the various croppings revealed that small differences in the cropped areas, represented by croppings 1 vs. 4, did not affect significantly the outcomes (Wilcoxon signed rank test, $p > 0.01$). On the contrary, larger variations in the cropped surfaces, such as those between Crops 1 vs. 2, led to significant differences in the distances between the measurement areas ($p < 0.01$, Fig 5). These findings suggest that if the operator is interested in assessing the overall asymmetry of the face as a single module, entire facial images should be used for valid outcomes. Otherwise, the outcomes should be interpreted according to the extent of the superimposed facial surface models used each time. The less extended surface models consistently showed smaller amount of asymmetries, which is expected due to the better best-fit approximation of the measurement areas on these. Thus, the interpretation of the findings should always take into account the extent of the superimposed facial model as compared to the area of interest. Visualization of the colour-coded distance maps confirmed these findings on an individual basis, suggesting minor differences between Crops 1 and 4, as well as minor differences between Crops 2 and 3. On the contrary, there were considerable differences of Crops 1 and 4 with Crops 2 and 3, with differences in individual cases and areas often reaching 2 mm. The similar outcomes provided by Crops 2 and 3 indicate that even models with unilaterally missing structures can provide valid outcomes, but the interpretation should consider only the areas that are present on both sides (Fig 6, S1–S3 Figs).

### Effect of differential facial surface model extent on midsagittal plane definition

The different croppings did not affect significantly the subsequently generated midsagittal planes, with the average differences between them being consistently small and almost always not statistically significant (Friedman test, $p > 0.05$, Fig 7).

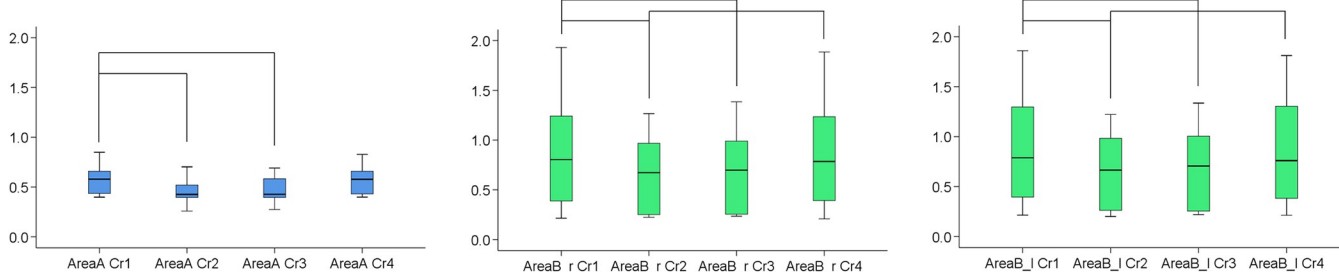

**Fig 5. Asymmetry measurements on differently cropped models.** Box plots showing the MAD (mm) between superimposed original and mirrored models that were cropped in different ways, at the different areas tested (Blue: Area A, Green: Area B right and Area B left). The lines connecting the different boxes indicate statistically significant differences between them (p<0.01). Cr: Crop, r: right, l: left.

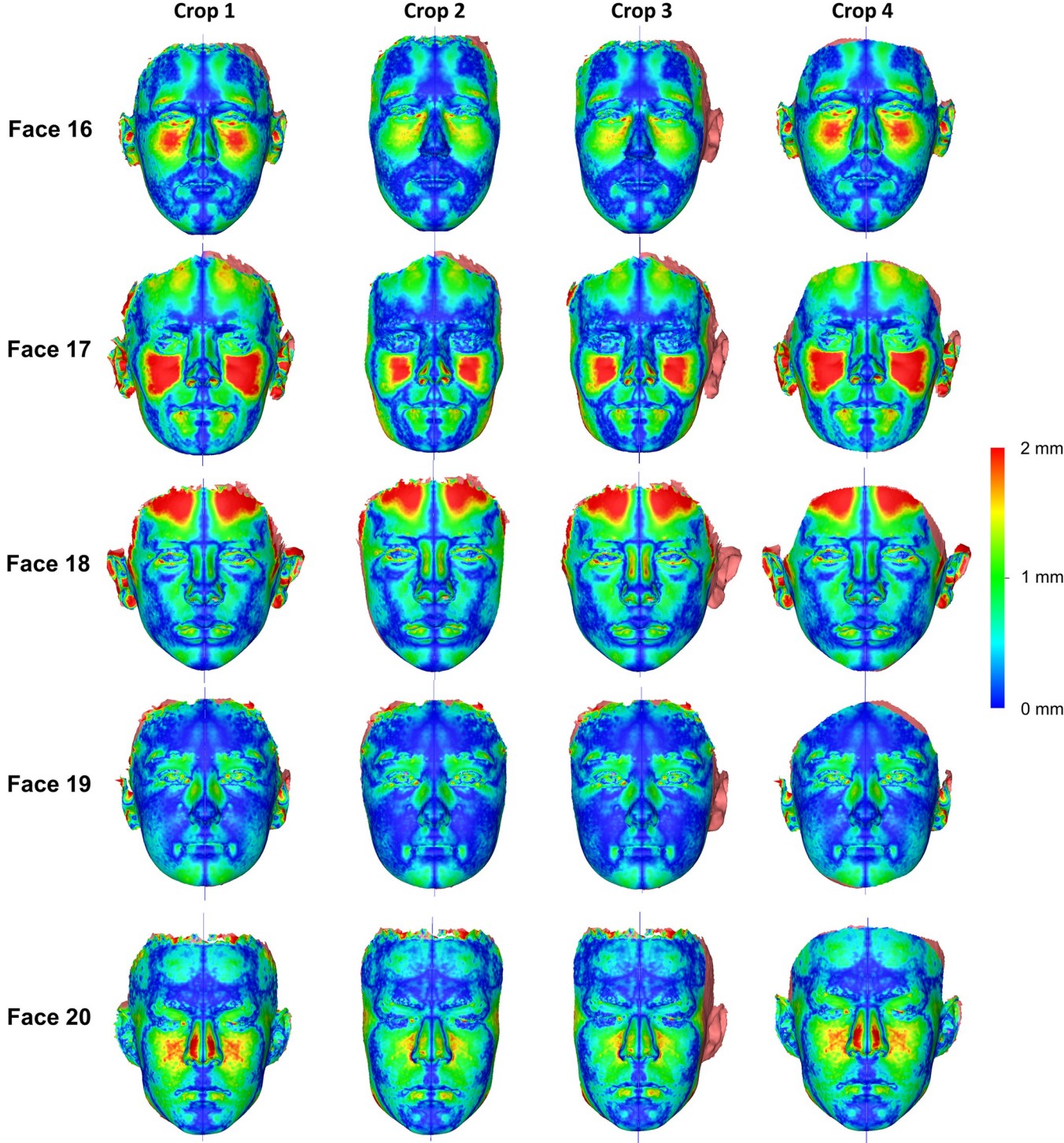

**Fig 6. Colour-coded distance maps showing asymmetries on differently cropped facial surface models.** These were generated through best-fit approximation of the surface models of five individuals with their mirrored duplicates (Faces 15–20).

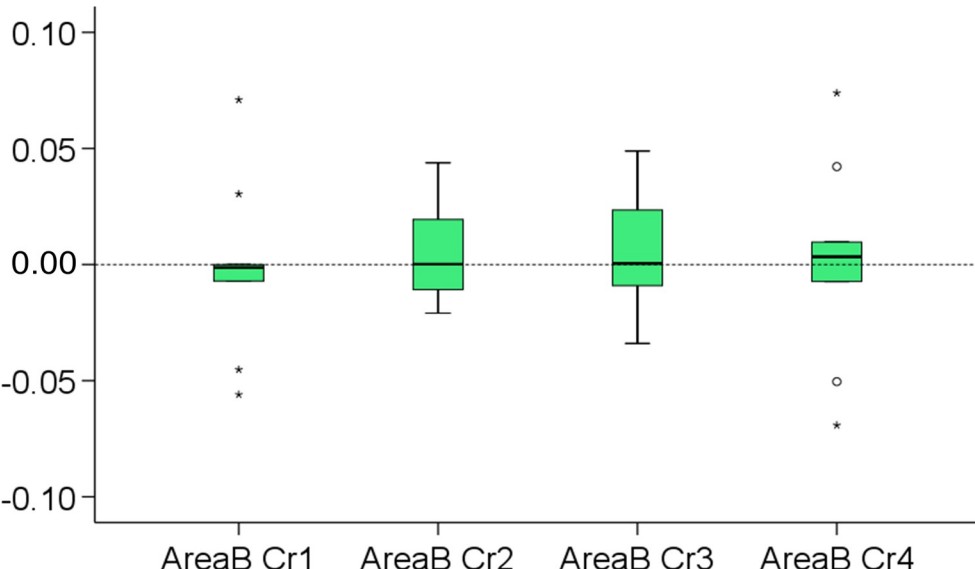

**Fig 7. Differences of the midsagittal planes generated following superimposition of differently cropped models.** Box plots showing the differences of the midsagittal planes generated following superimposition of Crops 2, 3 and 4 with their mirrored duplicates, from that generated from the superimposition of Crop 1 with its mirrored duplicate (reference) (Z: lateral movement in mm; Xrot: rotation around the anteroposterior axis in ˚; Yrot: rotation around the vertical axis in ˚). Outliers are shown as black circles. The lines connecting the different boxes indicate statistically significant differences between them (p<0.01). rot: rotation, Cr: Crop.

## Validation of the superimposition of original with mirrored models

The two operators consistently judged that all coloured coded distance maps constructed to assess facial asymmetry showed similar distances of the superimposed models in corresponding anatomical areas at the two contralateral sides (right/left). These appeared in the colour maps as right/left symmetric colour patterns towards the midsagittal plane.

The aforementioned qualitative assessment was confirmed quantitatively since differences in MAD between corresponding bilateral measurement areas (Area B right vs. Area B left) were consistently negligible (< 0.1 mm) and not significantly different between the various croppings (Friedman test, p > 0.05, Fig 8).

## Intra- and inter-operator method error

The differences between MADs of repeatedly superimposed mirrored and original models and between repeatedly created midsagittal planes were negligible for all croppings and for all cases

**Fig 8. Differences in asymmetry values between corresponding bilateral measurement areas.** Box plots showing the differences in the MAD (mm) of the right from the left (mirrored) measurement area, in each pair of the differently cropped, superimposed, original and mirrored models. Outliers are shown as black circles or stars in more extreme cases.

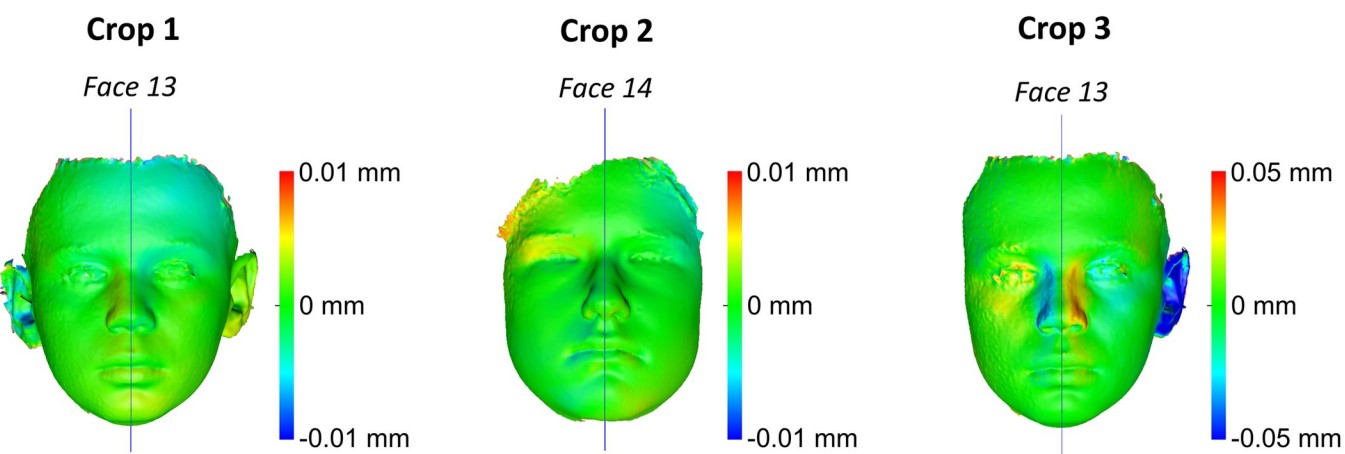

**Fig 9. Maximum intra-operator error in each Cropping group.** Colour-coded distance maps showing the differences between repeatedly superimposed mirrored models and the subsequently generated midsagittal planes, for the cases that showed the maximum intra-operator error (highest MAD values in Area A + Area B right + Area B left) in each Cropping group.

tested, indicating perfect intra- and inter-operator reproducibility of the applied methods (S4–S6 Figs). One sample t-test for the differences between MADs of repeatedly superimposed mirrored and original models and for repeatedly created midsagittal planes showed consistently p values higher than 0.05, both for intra- and inter-operator error. Fig 9 shows colour-coded distance maps of repeatedly superimposed mirrored models on the original model for the cases that showed the maximum intra-operator error in each Cropping group. Fig 10 shows colour-coded distance maps of repeatedly superimposed mirrored models on the original model for the cases that showed the maximum inter-operator error in each Cropping group. In all tested cases the observed differences between repeated superimpositions are negligible (< 0.05 mm), apart from Crop 4, which showed differences up to 0.5 mm. However, in Crop 4 the repeatedly superimposed facial models were not identical, as in all other cases, since they were each time individually cropped by the operator.

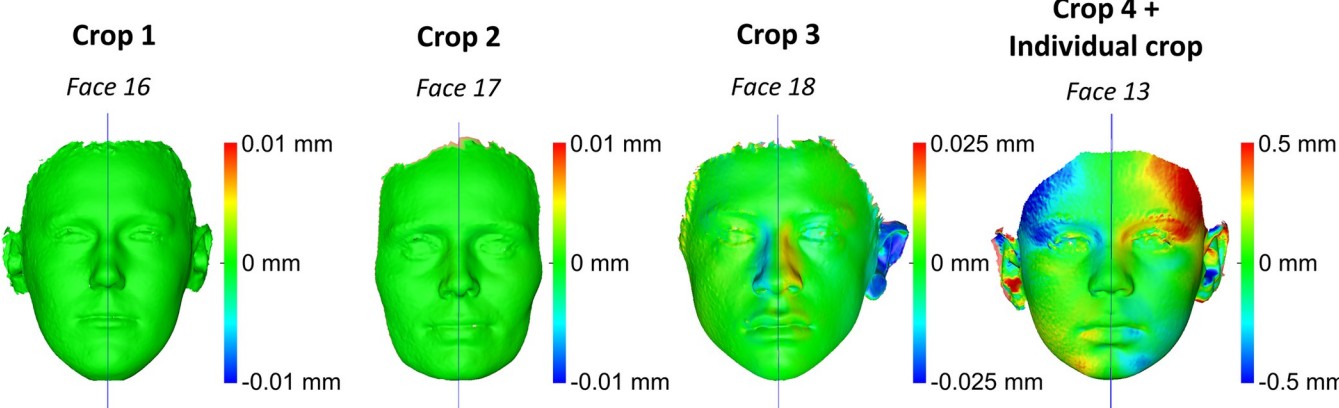

**Fig 10. Maximum inter-operator error in each Cropping group.** Colour-coded distance maps showing the differences between repeatedly superimposed mirrored models and the subsequently generated midsagittal planes, for the cases that showed the maximum inter-operator error (highest MAD values in Area A + Area B right + Area B left) in each Cropping group.

**Table 1. Effects of operator differences in superimposition outcomes on the generated midsagittal planes.**

|  | Crop 1 | Crop 2 | Crop 3 | Crop 4 |
|---|---|---|---|---|
|  | r[a] (p) | r[a] (p) | r[a] (p) | r[a] (p) |
| Z movement (mm) | 0.38 (0.276) | -0.55 (0.098) | -0.76 (0.011)* | -0.01 (0.987) |
| X rotation (˚) | 0.39 (0.260) | -0.81 (0.005)* | -0.72 (0.019)* | -0.55 (0.098) |
| Y rotation (˚) | 0.92 (0.000)* | -0.58 (0.082) | -0.82 (0.004)* | -0.43 (0.214) |

[a]Spearman's correlations indicating the effects of differences between operators in superimposition outcomes (MAD values in Area A + Area B right + Area B left), on the generated midsagittal planes (n = 10).

*$p < 0.05$

### Effect of inter-operator superimposition error on midsagittal plane definition

Spearman's correlations indicated that in certain cases, the higher the differences between operators in superimposition outcomes (MAD values in Area A + Area B right + Area B left), the higher the differences in the generated midsagittal planes (n = 10). This was mostly evident for Crop 3 (Table 1), probably because it showed relatively higher errors. On the other hand, the inter-operator error for Crop 1 and 2 was negligible (Fig 10, S5 Fig), whereas in Crop 4, apart from the superimposition error, the different cropping by each operator might have skewed the respective outcome.

## Discussion

Assessment of facial asymmetry is an important outcome for many different disciplines, since it has social, clinical, developmental and evolutionary implications. For example, facial appearance is a crucial component of physical attractiveness, which affects several aspects of personal, social, and professional life. Midsagittal plane determination is also crucial for all biological, clinical and anthropological fields that perform facial analyses [2, 5, 20]. The present study suggested and thoroughly validated an efficient method for 3D facial surface symmetry assessment and bias-free, automated midsagittal plane definition. The method proved valid and highly reproducible for different extents of facial surface models, providing that the interpretation of asymmetry outcomes is performed in regard to the superimposition reference area selected each time. On the other hand, the midsagittal plane definition was robust despite the differences in the extent of the surface models and the bilateral presence of all anatomical areas.

The suggested method could be considered superior to existing methods for various reasons. It is objective (not prone to operator error), since it is automated, landmark-free and does not require arbitrary decisions, such as the definition of the middle of a 3D object by weighing certain anatomical points or areas more than others. It is also very efficient and easy to apply and understand. A whole-face colour-coded distance map requires very few steps, takes less than a minute to compute and provides a detailed assessment of the entire tested object. Quantification of asymmetry at any area of interest can be easily performed by selecting the area and exporting the distances between the two mesh surfaces (original and mirrored) at each mesh vertex. The method can be broadly applied, since its software implementation is straightforward and 3D facial surface models are becoming readily available. Such models are most often acquired through direct facial surface scans performed either with specialized scanner or camera equipment or, nowadays, even with high-end mobile phones [31]. In addition to direct 3D facial images, facial surface models can be derived from 3D radiographic volumes (CT or CBCT images), if these are available for other purposes [32]. It has been shown that

even the lower radiation CBCT derived facial soft-tissue surface models are quite similar to those obtained by stereophotogrammetry [33–35], thus offering adequate quality for most applications.

There are several reports in the literature concerning 3D facial asymmetry assessment methods. Recently, Lum et al. (2020) [36] presented a dense correspondence method applied to 3D facial surface models. The idea was to apply a landmark free-approach, which is similar to ours, but the suggested process involved additional steps to obtain contralateral corresponding points on each surface and dense correspondence between the tested samples, which is time consuming and adds complexity to the process. Furthermore, the required transformation of the original surface models might have introduced errors. Similar approaches, but sharing the same limitations, have been suggested by other researchers [24, 37, 38]. Taylor et al. (2014) [39] proposed a similar method, aligning the original and reflected surfaces by "maximizing the fit" between them. It is not clear how this was achieved; the Procrustes method was mentioned but this requires the establishment of correspondences between the two surfaces. Also, Procrustes alignment may lead to errors because it takes all mesh vertices into account, even those that should be ignored, e.g. in cases of asymmetric cropping. There are previous reports in the literature that performed best fit approximations of originally obtained surface models to their mirrored equivalents to assess facial asymmetry, as well as researchers that constructed midsagittal symmetry planes in a similar manner as we do here [40–43]. A justification of these approaches has been presented by Benz et al., (2002) [41], but a comprehensive validation, robustness assessment, and proof of the concept is provided here.

In the present study, the measurement area selection and croppings A, B, and C were performed only once and used for several purposes to eliminate confounding due to these factors. The area selection needs to be identical to have directly comparable outcomes and the extent of the facial surface used for the best fit superimposition was shown to affect the outcomes. The effect of slight differences in facial surface cropping on the outcomes was tested through cropping D, which was each time individually performed by each operator. A prerequisite for optimal asymmetry assessment through the present 3D superimposition method is the best possible approximation of the original with the mirrored model. Since the superimposed surface models are identical, but mirrored in one axis, this, by definition can be achieved only through setting the estimated overlap of meshes to 100%. This was confirmed by exploratory testing on random samples. For this reason we did not test here systematically any reduced estimated overlap values, as we did previously in other studies [44–46]. The same prerequisite holds true for the validity of the resulting midsagittal plane. We showed here that for any orientation between an original and its mirrored model, the midpoints of the lines connecting bilateral corresponding points of the two models are coplanar and they can be considered the "mirror" (Fig 3). Therefore, a "mirror" can be drawn between any original and its mirrored model, but if this mirror represents that midsagittal plane, depends on the performance of the ICP. In the present study, the latter has been assessed qualitatively through the viewing of the resulting superimposed models and of the respective colour-coded distance maps and quantitatively through the comparison of the distances of the closest points between the original and the flipped model, at two identical but contralateral areas (named here Area B right and Area B left). Regarding the qualitative assessment, two operators assessed all images independently and agreed on no visually detectable irregularities. For the quantitative assessments, if the method worked properly, the subtraction of the MAD of the two models (original and mirrored) at one area from that of its contralateral area should equal zero and this was confirmed in the study (Fig 8).

Through the application of the present method, the operator can easily assess the overall facial asymmetry in 3D using colour-coded distance maps. Though this is an automated

approach, not depended on bias-generating operator decisions, its major limitation is that the distances shown in the colour maps are not between points with strict anatomical correspondence, but between closest points. It can be argued that the validation process applied here confirmed, to a certain level, the anatomical correspondence between the approximated areas. For absolute correspondence, anatomical landmarks can be placed on the surface models and asymmetries can be calculated by comparing the distances of the original to the mirrored landmark configurations [5]. This solves the correspondence issue, but introduces limitations related to landmark identification. Furthermore, to achieve similar level of information, hundreds, if not thousands of landmarks, need to be placed; a process requiring considerable amount of time and expertise. Another option would be to apply dense correspondence methods, as those described above, on the original surface models, prior to the application of the current method. However, this would add complexity and might introduce errors.

With the current method, following a perfect best fit approximation of the original and the mirrored facial surface, the software calculates a plane located at the middle of each original point with its mirrored equivalent. This plane is not affected or defined by any arbitrarily selected landmarks. Given that the original and the mirrored facial surface models are best fit approximated this is the true, geometrically defined midsagittal plane of the face. In case that anatomical correspondence needs to be guaranteed, after the bias-free definition of the midsagittal plane, landmarks can be used at a second stage e.g. to measure distances between anatomically corresponding contralateral points from the midsagittal plane, when comparing the two halves of the face. In such cases, the unbiased construction of the midsagittal plane (geometrically and not arbitrarily defined by any operator or landmark selection) could provide a viable solution to the anatomical correspondence problem, with the errors deriving solely from the landmark identification of the assessment points. Nevertheless, it remains questionable if the latter approach, with the associated identification error, provides better correspondence than that of the proposed closest point approximation analysis, which is far more cost and time-efficient. Landmark identification error in 3D facial soft-tissue surface models has been reported to often exceed 0.5 mm on average [47, 48].

The basic principles underlying the present method, and thus, the method itself, are expected to be valid for broad applications. The method could be applicable to any object or biological form, but this remains to be tested.

## Conclusions

The present study thoroughly tested and validated an efficient, automated, and bias-free method to assess 3D facial surface symmetry and construct the midsagittal plane. The method was applicable to different extents of facial surface models, providing that the outcome interpretation is performed in regard to the selected superimposition reference area. The midsagittal plane was not affected by the extent of the surface models and the bilateral presence of all anatomical areas.

## Supporting information

**S1 Dataset. Anonymized dataset including all data analysed in the study.**
(XLSX)

**S1 Video. Application of the 3D model superimposition and midsagittal plane construction method in actual conditions.**
(MP4)

**S1 Fig. Colour-coded distance maps showing asymmetries on differently cropped facial surface models.** These were generated through best-fit approximation of the surface models of five individuals with their mirrored duplicates (Faces 1–5).
(TIF)

**S2 Fig. Colour-coded distance maps showing asymmetries on differently cropped facial surface models.** These were generated through best-fit approximation of the surface models of five individuals with their mirrored duplicates (Faces -10).
(TIF)

**S3 Fig. Colour-coded distance maps showing asymmetries on differently cropped facial surface models.** These were generated through best-fit approximation of the surface models of five individuals with their mirrored duplicates (Faces 10–15).
(TIF)

**S4 Fig. Intra-operator error in asymmetry assessment and in midsagittal plane generation.** Box plots showing the intra-operator error in asymmetry assessments (upper and lowest row) and in midsagittal plane generation (middle row; Z: lateral movement in mm; Xrot: rotation around the anteroposterior axis in˚; Yrot: rotation around the vertical axis in˚). Outliers are shown as black circles or stars in more extreme cases. rot: rotation, Cr: Crop.
(TIF)

**S5 Fig. Inter-operator error in asymmetry assessment and in midsagittal plane generation.** Box plots showing the inter-operator error in asymmetry assessments (upper and lowest row) and in midsagittal plane generation (middle row; Z: lateral movement in mm; Xrot: rotation around the anteroposterior axis in˚; Yrot: rotation around the vertical axis in˚). Outliers are shown as black circles or stars in more extreme cases. rot: rotation, Cr: Crop.
(TIF)

**S6 Fig. Inter-operator error with additional individual cropping effect.** Box plots showing the inter-operator error in asymmetry assessments (left side, Friedman test: $p > 0.05$) and in midsagittal plane generation (right side; Z: lateral movement in mm; Xrot: rotation around the anteroposterior axis in˚; Yrot: rotation around the vertical axis in˚; one sample t-test: $p > 0.05$) with Crop 4, where each facial surface was individually cropped by each operator. Outliers are shown as black circles or stars in more extreme cases. rot: rotation, Cr: Crop.
(TIF)

**S1 Graphical abstract.**
(TIF)

## Acknowledgments

We are very grateful to Dr. Konstantinos Dritsas for the assistance in video editing.

## Author Contributions

**Conceptualization:** Nikolaos Gkantidis, Demetrios Halazonetis.

**Data curation:** Nikolaos Gkantidis, Jasmina Opacic.

**Formal analysis:** Nikolaos Gkantidis, Jasmina Opacic, Georgios Kanavakis.

**Funding acquisition:** Demetrios Halazonetis.

**Methodology:** Nikolaos Gkantidis.

**Project administration:** Nikolaos Gkantidis.

**Resources:** Christos Katsaros.

**Software:** Demetrios Halazonetis.

**Supervision:** Nikolaos Gkantidis, Demetrios Halazonetis.

**Validation:** Nikolaos Gkantidis.

**Visualization:** Nikolaos Gkantidis, Jasmina Opacic, Demetrios Halazonetis.

**Writing – original draft:** Nikolaos Gkantidis, Jasmina Opacic.

**Writing – review & editing:** Nikolaos Gkantidis, Georgios Kanavakis, Christos Katsaros, Demetrios Halazonetis.

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
