## [Decision Letter · Decision Letter 0]

3 Nov 2023

Facial asymmetry and midsagittal plane definition in 3D: a bias-free, automated method

PONE-D-23-27813

Dear Dr. Gkantidis,

We’re pleased to inform you that your manuscript has been judged scientifically suitable for publication and will be formally accepted for publication once it meets all outstanding technical requirements.

Kind regards,

Johari Yap Abdullah, B.S. & I.T, GradDip ICT, M.Sc, Ph.D.

Academic Editor

PLOS ONE

I have read the journal's policy and the authors of this manuscript have the following competing interests: Demetrios Halazonetis owns stock in dHAL Software, the company that markets Viewbox 4. Demetrios Halazonetis was not involved in data generation and analysis, and thus, could not affect the study outcomes. All other authors declare no known competing financial interests or personal relationships that could have appeared to influence the work reported in this paper.   

We note that one or more of the authors are employed by a commercial company. 

“The funder provided support in the form of salaries for authors, but did not have any additional role in the study design, data collection and analysis, decision to publish, or preparation of the manuscript. The specific roles of these authors are articulated in the ‘author contributions’ section.”

Please respond by return email with an updated Funding Statement and Competing Interests Statement and we will change the online submission form on your behalf.

Reviewers' comments:

Reviewer's Responses to Questions

**Comments to the Author**

1. Is the manuscript technically sound, and do the data support the conclusions?

Reviewer #1: Partly

Reviewer #2: Yes

2. Has the statistical analysis been performed appropriately and rigorously? 

Reviewer #1: N/A

Reviewer #2: Yes

3. Have the authors made all data underlying the findings in their manuscript fully available?

Reviewer #1: Yes

Reviewer #2: Yes

4. Is the manuscript presented in an intelligible fashion and written in standard English?

Reviewer #1: Yes

Reviewer #2: Yes

5. Review Comments to the Author

Reviewer #1: The study presents a solution based on automatic tools with a good approach to the software used, the limitations of the approach are commented and solutions provided in the study itself. The other problems related to difficulties seem more inherent to the structural nature of the face than to the limitations of the tools used on the study.

Reviewer #2: Dear Authors,

Thank you for the article. It was interesting to read it. The methodology and the flow of the article was easily understandable. The article can be published in its current form. Provided there are no queries from other reviewers.

6. PLOS authors have the option to publish the peer review history of their article (what does this mean?). If published, this will include your full peer review and any attached files.

Reviewer #1: No

Reviewer #2: No

---

## [Editor Report · Acceptance letter]

14 Nov 2023

PONE-D-23-27813 

Facial asymmetry and midsagittal plane definition in 3D: a bias-free, automated method 

Dear Dr. Gkantidis:

I'm pleased to inform you that your manuscript has been deemed suitable for publication in PLOS ONE. Congratulations! Your manuscript is now with our production department. 

Kind regards, 

on behalf of

Dr. Johari Yap Abdullah 

Academic Editor

PLOS ONE